# SLC35G3 is a UDP-N-acetylglucosamine transporter for sperm glycoprotein formation and underpins male fertility in mice

Daisuke Mashiko[1,2], Shingo Tonai[1], Haruhiko Miyata[1], Martin M Matzuk[3,4], Masahito Ikawa[1,2,5,6,7]*

[1]Department of Experimental Genome Research, Research Institute for Microbial Diseases, The University of Osaka, Osaka, Japan; [2]Immunology Frontier Research Center, The University of Osaka, Osaka, Japan; [3]Department of Pathology and Immunology, Baylor College of Medicine, Houston, United States; [4]Center for Drug Discovery, Baylor College of Medicine, Houston, United States; [5]Center for Advanced Modalities and Drug Delivery System, The University of Osaka, Osaka, Japan; [6]Center for Infectious Disease Education and Research, The University of Osaka, Osaka, Japan; [7]The Institute of Medical Science, The University of Tokyo, Tokyo, Japan

*For correspondence:
ikawa@biken.osaka-u.ac.jp

## eLife Assessment

This **valuable** study reports the physiological function of a putative transmembrane UDP-N-acetylglucosamine transporter called SLC35G3 in spermatogenesis. The conclusion that SLC35G3 is a new and essential factor for male fertility in mice and probably in humans is supported by **convincing** data. This study will be of interest to reproductive biologists and physicians working on male infertility.

**Abstract** Despite the recognized importance of glycans in biological phenomena, their complex roles in spermatogenesis and sperm function remain unclear. SLC35G3, a 10-transmembrane protein specifically found in early round spermatids, belongs to the sugar-nucleotide transporter family, indicating its involvement in glycan formation. In this study, we found that *Slc35g3* knockout male mice were sterile due to impaired sperm functions in uterotubal junction passage, zona pellucida binding, and oocyte fusion. Mouse SLC35G3 has UDP-GlcNAc transporter activity, and its ablation caused abnormal processing of the sperm plasma membrane and acrosome membrane proteins. Reported human *SLC35G3* mutations (F267L and T179HfsTer27) diminished the UDP-GlcNAc transporter activity of SLC35G3, implying infertility risks in males carrying these mutations. Our findings unveil the vital roles of SLC35G3 in the glycan formation of sperm membrane proteins critical for sperm-fertilizing ability.

## Introduction

Glycosylation is a post-translational modification that ensures target protein synthesis, secretion, stability, characterization, and/or function (*Matzuk and Boime, 1988*; *Matzuk et al., 1989*; *Schjoldager et al., 2020*). In the endoplasmic reticulum (ER), the oligosaccharyltransferase complex (OSTC) co-translationally transfers core glycans assembled on dolichol phosphate to asparagine

residues of nascent proteins. Subsequently, these proteins undergo a quality control process in which the core glycan structure is processed by glucosidases, resulting in a monoglucosylated form that binds to ER lectin chaperones, calnexin (CANX) and calreticulin (CALR). Once disulfide bonds are correctly formed by protein disulfide isomerase (PDI) and the protein is properly folded, the protein is transported to the Golgi (*Tannous et al., 2015*; *Ikawa et al., 1997*). Notably, there are testis-specific proteins that are required for these processes and the regulation of sperm-fertilizing ability. Recent studies suggest that FREY tightly interacts with proteins involved in N-glycosylation, and its disruption destabilizes OSTC and causes subsequent ablation of the acrosomal membrane proteins essential for sperm-egg fusion (*Contreras et al., 2022*; *Lu et al., 2023*). In addition to the CANX/CALR/PDI complex in somatic cells, their testis-specific paralogs, CLGN/CALR3/PDILT, are required for ADAM3 sperm membrane glycoprotein maturation to equip sperm with fertilization competence, including the ability to pass through the uterotubal junction (UTJ) (*Tokuhiro et al., 2012*).

In the ER to Golgi secretory pathway, more than 200 glycosyltransferases, such as mannosyl (alpha-1,3-)-glycoprotein beta-1,2-N-acetylglucosaminyltransferase (MGAT) and N-acetylgalactosaminyl-transferase (GALNT), add further diversity by conferring various properties, such as solubility and adhesiveness to the proteins. Once glycoproteins reach the cell surface, some are secreted to form the extracellular matrix, while others remain and contribute to cell adhesion and interactions with substrates or other cells. Among these glycosyltransferase-like proteins, DPY19L2 (a probable C-mannosyltransferase), MGAT4D, MGAT4E, MGAT4F, and GALNTL5 show testis-specific expression by in silico analysis (*Oura et al., 2022*). *Dpy19l2* knockout mice are infertile due to globozoospermia (*Pierre et al., 2012*), and a mutation in *Galntl5* resulted in asthenozoospermia (*Takasaki et al., 2014*). Of note, GALNTL5 does not exhibit transferase activity in vitro (*Lu et al., 2019*). While *Mgat4d* knockout mice are fertile (*Raman et al., 2012*), *Mgat4e* and *Mgat4f* orthologs do not exist in humans, and their knockout mice need to be generated to reveal if they have indispensable or redundant functions in mice. Collectively, these findings suggest that the spermatogenic cells have a unique system for the production and quality control of glycoproteins, and some of them are critical for spermatogenesis, sperm functions, and male fertility.

In the present study, we focused on the solute carrier (SLC) 35 family of nucleotide sugar transporters, which are responsible for importing sugars that serve as substrates for glycosyltransferases. Sugars are conjugated to nucleotides and transported by specific SLC35 family antiporters into the ER and Golgi apparatus, where glycosyltransferases utilize them to modify target proteins. Glycan structures are synthesized from sugars including D-glucose (Glc), D-galactose (Gal), N-acetyl-D-glucosamine (GlcNAc), N-acetyl-D-galactosamine (GalNAc), L-fucose (Fuc), D-glucuronic acid (GlcA), D-mannose (Man), N-acetylneuraminic acid (Neu), and D-xylose (Xyl). Among the SLC35 paralogs, SLC35A1 transports CMP-Sialic Acid, SLC35A2 transports UDP-Gal, SLC35B4 transports UDP-GlcNAc, and SLC35C1 transports GDP-Fuc (*Ahuja and Whorton, 2019*). Of the 27 SLC35 family members, most show ubiquitous expressions, including in spermatogenic cells. Notably, in silico analysis revealed that *Slc35g3* is the only SLC35 family member specifically expressed in the testis. *Slc35g3* emerged in amphibians and is conserved in primates. We elucidated the biochemical properties of SLC35G3 in vitro and generated *Slc35g3* knockout mice to study its physiological functions in vivo. We discovered that SLC35G3 is a spermatogenic cell-specific UDP-GlcNAc transporter, and *Slc35g3* ablation results in abnormal processing of sperm plasma membrane and acrosome membrane glycoproteins required for sperm-fertilizing ability and male fertility.

## Results

### SLC35G3 is expressed during late spermatogenesis and localized in the Golgi apparatus

In mice, *Slc35g3* comprises two coding exons and is located on chromosome 11, whereas it is located on chromosome 17 in humans. The TreeFam (*Ruan et al., 2008*) data confirmed the evolutionary conservation of *Slc35g3* among vertebrates (*Figure 1A*). RT-PCR analysis indicates that it is prominently expressed in the testis, beginning 21 days postpartum (*Figure 1B*), suggesting that its expression begins in round spermatids in mice. The Mammalian Reproductive Genetics Database (*Robertson et al., 2020*) revealed that *Slc35g3* is the only mouse SLC35 family that shows a testis-specific transcription pattern (*Figure 1—figure supplement 1*). A previous scRNA-seq analysis suggested that

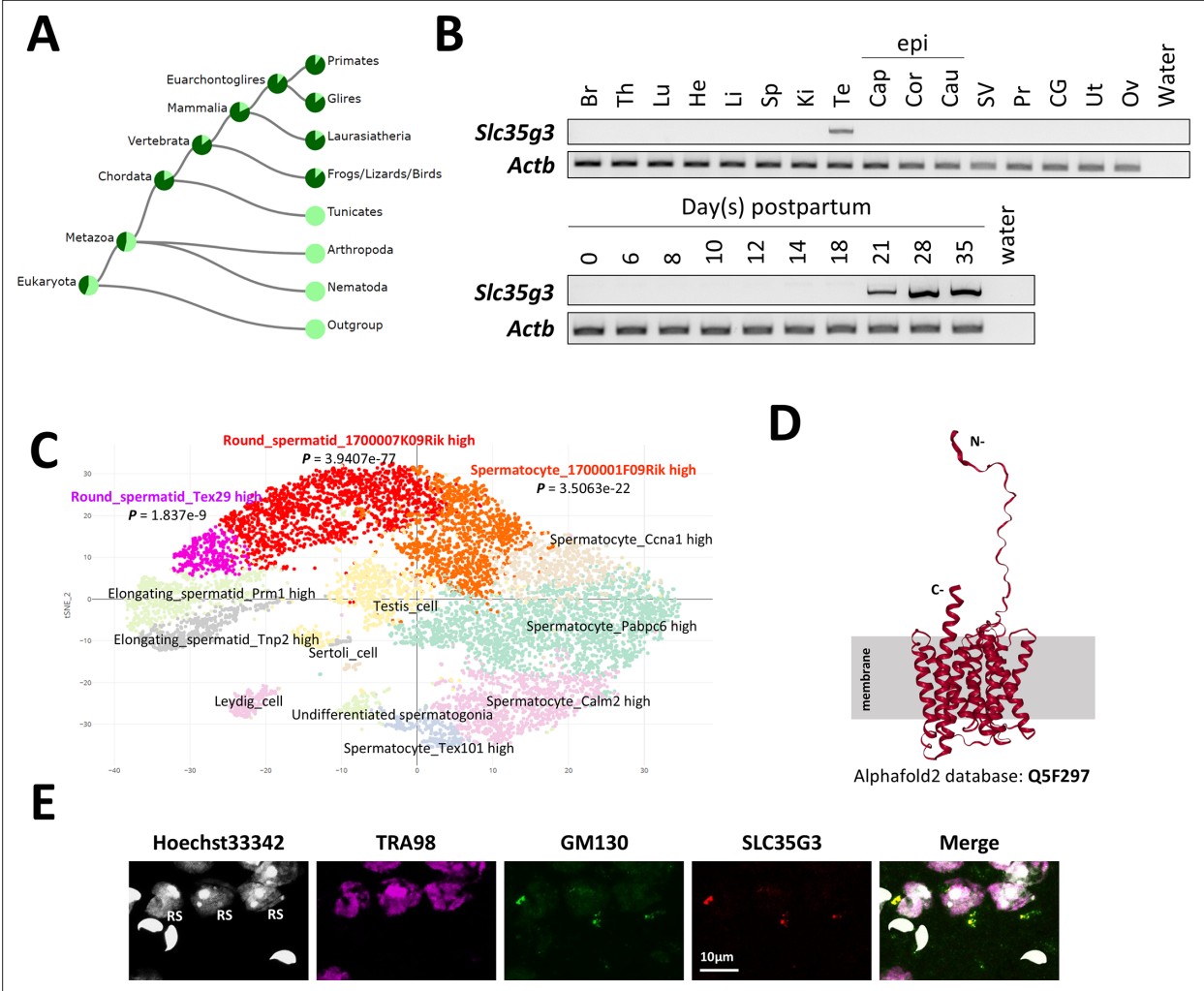

**Figure 1.** SLC35G3 is a multi-pass transmembrane protein with unique testes-specific expression in the Golgi apparatus during early spermiogenesis. (**A**) Phylogenetic tree of *Slc35g3* from the TreeFam database, with dark green areas indicating the presence and light green areas indicating the absence of *Slc35g3*. (**B**) RT-PCR results across multiple tissues (upper panel) and from testes at various days postpartum (lower panel); Br: brain, Th: thymus, Lu: lung, He: heart, Li: liver, Sp: spleen, Ki: kidney, Te: testis; Epi: epididymis, Cap: caput epididymis, Cor: corpus epididymis, Cau: cauda epididymis; SV: seminal vesicle, Pr: prostate, CG: coagulating gland, Ut: uterus, Ov: ovary. β-actin (Actb) was used as the loading control. (**C**) scRNA-seq prediction of cells strongly expressing *Slc35g3* mRNA (Mouse Cell Atlas). Dots with low transparency represent cells with predicted expression. (**D**) SLC35G3 structure predicted using AlphaFold. (**E**) From left to right: Hoechst33342 staining image, SLC35G3 immunostaining image, GM130 immunostaining image, and merged image of wild-type testicular germ cells. Scale bar: 10 μm, RS: round spermatid.

The online version of this article includes the following source data and figure supplement(s) for figure 1:

**Source data 1.** Gel for *Figure 1*, indicating the relevant bands.

**Source data 2.** Original files for gel analysis displayed in *Figure 1*.

**Figure supplement 1.** *Slc35g3* is the sole Slc35 family member that is testis-specific, with highest mRNA levels in round spermatids.

**Figure supplement 2.** SLC35G3 has the potential to form a homodimer.

**Figure supplement 2—source data 1.** Gel for *Figure 1—figure supplement 2*, indicating the relevant bands.

**Figure supplement 2—source data 2.** Original files for gel analysis displayed in *Figure 1—figure supplement 2*.

transcription of *Slc35g3* initiates in round spermatids (*Figure 1C*; Mouse Cell Atlas; 17). Both Alpha-Fold2 (*Jumper et al., 2021*) and TOPCONS (*Tsirigos et al., 2015*) analyses supported that SLC35G3 has 10 transmembrane domains (*Figure 1D*), likely forming a homodimer (*Figure 1—figure supplement 2*). Immunostaining colocalizes SLC35G3 with Golgi marker (GM) 130 (Golgin A2), indicating SLC35G3 localization in the Golgi of mouse testicular germ cells (*Figure 1E*).

## Slc35g3⁻/⁻ mice showed male infertility

To investigate the roles of *Slc35g3* in male reproduction, we used CRISPR/Cas9 to generate a homozygous knockout mouse line (*Slc35g3⁻/⁻*) with an 1804 bp deletion on a hybrid B6D2 background. This deletion resulted in the loss of the entire *Slc35g3* coding region (**Figure 2A and B**), indicating that it should be a null allele.

Slc35g3⁻/⁻ mice exhibit grossly normal development, appearance, and behavior, consistent with its testis-restricted expression. Absence of the *Slc35g3* mRNA and SLC35G3 protein in the *Slc35g3⁻/⁻* testes was verified by RNA-seq (**Figure 2—figure supplement 1**) and western blot analysis (**Figure 2C**), respectively. The specific expression of SLC35G3 in the testis, but not in epididymal sperm, suggests that its function is restricted to spermatogenesis. Moreover, immunofluorescence of SLC35G3 confirmed its absence in the Golgi of *Slc35g3⁻/⁻* mice (**Figure 2—figure supplement 2**). Testes of *Slc35g3⁻/⁻* male mice appeared normal in both appearance and weight (**Figure 2D and E**; +/+ vs. -/-, two-sided Student's *t*-test; p=0.42). Despite successful copulation, as evidenced by the presence of vaginal plugs, *Slc35g3⁻/⁻* male mice are sterile (**Figure 2F**; +/+ vs. -/-, two-sided Wilcoxon rank-sum test, p=2.87 × 10⁻¹⁰). Examination of seminiferous tubule and epididymis sections revealed no overt abnormalities (**Figure 2G** and **Figure 2—figure supplement 3**). Furthermore, computer-assisted sperm analysis revealed no significant differences in the motility of sperm from control and *Slc35g3⁻/⁻* males (**Figure 2—figure supplement 4**).

## Slc35g3⁻/⁻-derived sperm exhibit abnormal head morphology

Given the subtle morphological changes observed in *Slc35g3⁻/⁻*-derived sperm (**Figure 3A**), we employed elliptic Fourier descriptors (**Kuhl and Giardina, 1982**; **Mashiko et al., 2017**) to characterize the entire sperm head shape and conducted a principal component (PC) analysis (**Figure 3B, C and D**). Wild-type-derived sperm and *Slc35g3⁻/⁻*-derived sperm could be differentiated based on their PC2 analysis of the tip of the sperm heads, with *Slc35g3⁻/⁻*-derived sperm displaying a relatively higher PC2 value (**Figure 3C**), indicating the lack of the hook shape in *Slc35g3⁻/⁻*-derived sperm. The head shape of *Slc35g3⁻/⁻*-derived sperm resembled that of sperm from *Fam71f2⁻/⁻* (recently renamed as *Garin1a*, Golgi-associated RAB2 interactor 1A) (**Morohoshi et al., 2021**) mice (**Figure 3—figure supplement 1**). However, *Slc35g3⁻/⁻* mice exhibited a more severe fertility phenotype compared to *Fam71f2⁻/⁻* mice (average litter size = 0 and 4.4 pups/litter, respectively), suggesting that sperm head morphology is not the sole cause of sterility in *Slc35g3⁻/⁻* mice.

## Slc35g3⁻/⁻-derived sperm exhibit impaired zona pellucida (ZP) binding and fertilization

To further analyze the cause of infertility in *Slc35g3* null male mice, we performed an in vitro fertilization (IVF) assay. We first performed conventional IVF using cumulus-intact oocytes with 2×10⁵ sperm/mL insemination and found no oocytes fertilized with spermatozoa from *Slc35g3⁻/⁻* males (**Figure 4A**). By removing cumulus cells followed by insemination (**Figure 4B**), we found a decline in the number of *Slc35g3⁻/⁻*-derived spermatozoa bound to the ZP (**Figure 4C and D**), no oocytes fertilized as well (**Figure 4E**, Wilcoxon rank-sum test; p=0.0079). Further study, using ZP-free oocytes preloaded with Hoechst33342 (**Figure 4F**), revealed a significantly lower number of sperm fusing with the oocyte compared to control *Slc35g3⁺/⁻* (**Figure 4G and H**, +/- vs. -/-, Wilcoxon rank-sum test, p=1.71 × 10⁻²¹). Notably, oolemma fusion and fertilization were improved with a 10 times higher sperm concentration from *Slc35g3⁻/⁻* males but were still significantly decreased compared to the lower concentrations of sperm from controls (**Figure 4—figure supplement 1**). Lastly, IVF performed using cumulus-intact oocytes with 10 times more sperm insemination (2×10⁶ sperm/mL) resulted in a reduced fertilization rate (45.7%, 21/46), but we succeeded in obtaining 10 live pups from these fertilized eggs (**Figure 4—figure supplement 2**). Our results indicate that *Slc35g3⁻/⁻*-derived sperm have defects in ZP binding and oolemma fusion ability but produce viable offspring.

## Slc35g3-deficient mice show impaired sperm migration to the oviduct

As spermatozoa lacking ZP binding frequently cannot pass through the UTJ and reach the oviduct (**Tokuhiro et al., 2012**), we observed sperm UTJ passage after mating. This observation was facilitated by a red fluorescence signal in the sperm tails from Tg mice (CAG/*su9*-DsRed2, *Acr3*-eGFP) (**Hasuwa et al., 2010**; **Figure 5A**). Two hours after copulation with wild-type female mice (**Figure 5B**),

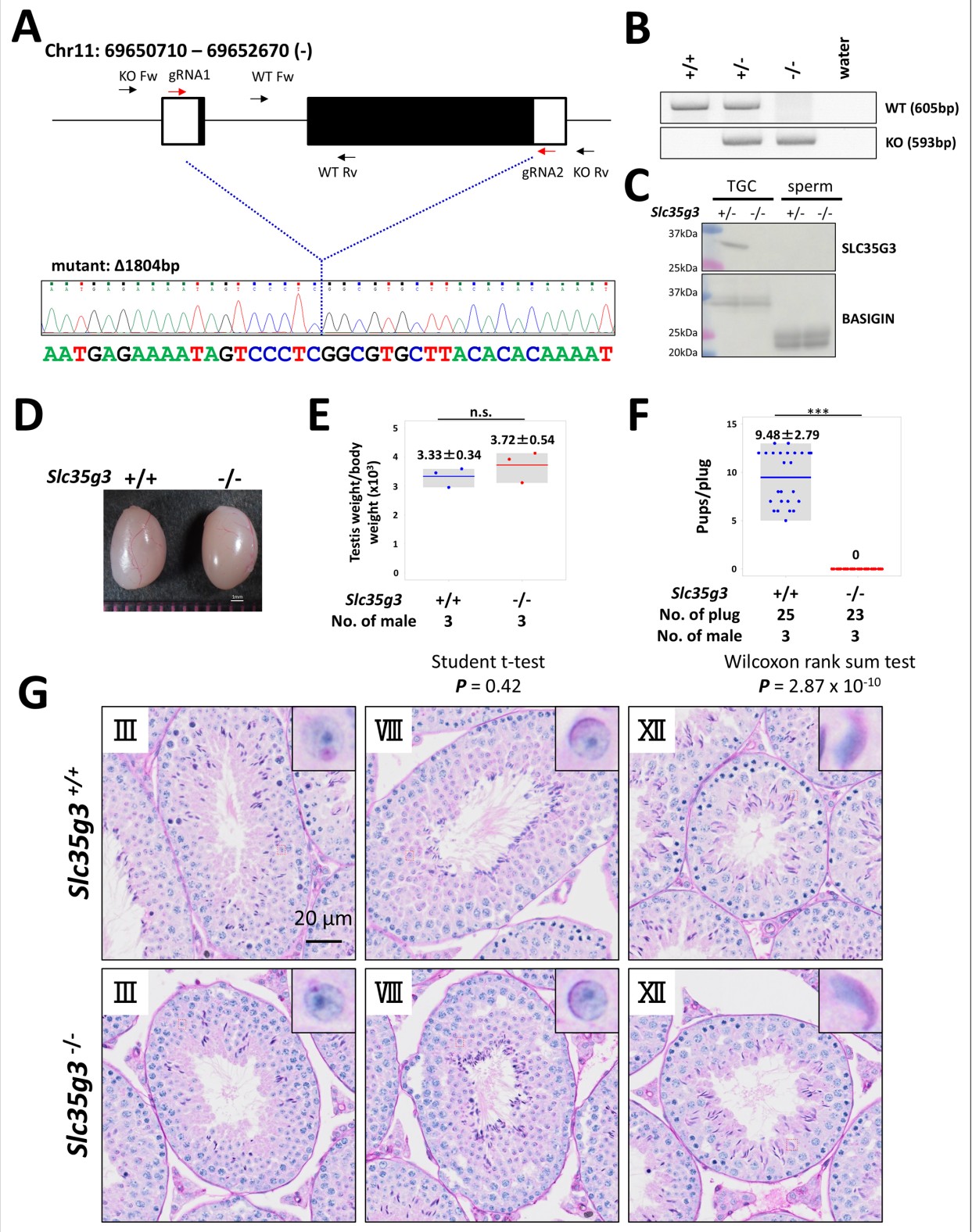

**Figure 2.** *Slc35g3*⁻/⁻ induces male sterility. (**A**) Depiction of *Slc35g3* gene location and structure, gRNA/primer design, and the sequencing result of the mutant (deleted) allele. (**B**) PCR genotyping results for *Slc35g3*⁺/⁺, *Slc35g3*⁺/⁻, *Slc35g3*⁻/⁻, and water are presented. (**C**) Western blot analysis results obtained with *Slc35g3*⁺/⁻ and *Slc35g3*⁻/⁻ testicular germ cells (TGC) lysates and *Slc35g3*⁺/⁻ and *Slc35g3*⁻/⁻-derived cauda epididymal sperm lysates are shown. (**D, E**) Similar testis sizes (**D**) and weights (**E**) from *Slc35g3*⁺/⁺ and *Slc35g3*⁻/⁻ mice (two-sided Student's *t*-test; p=0.42). (**F**) Comparison of the number of pups per vaginal plug between *Slc35g3*⁺/⁺ and *Slc35g3*⁻/⁻ mice (Wilcoxon rank-sum test; p=2.87 × 10⁻¹⁰). (**G**) Histological analysis of testis

*Figure 2 continued*

sections from *Slc35g3*+/+ mice (upper panels) and those from *Slc35g3*-/- mice (lower panels); images depict stages III (Golgi phase), VIII (acrosome phase), and XII (maturation phase).

The online version of this article includes the following source data and figure supplement(s) for figure 2:

**Source data 1.** Western blots and gel for *Figure 2*, indicating the relevant bands.

**Source data 2.** Original files for western blot and gel analysis displayed in *Figure 2*.

**Figure supplement 1.** Differentially expressed genes (DEG) analysis using testis RNA-seq data.

**Figure supplement 2.** SLC35G3 is localized in the Golgi apparatus, and the signal disappears in the knockout.

**Figure supplement 3.** Epididymis sections from *Slc35g3*+/+ and *Slc35g3*-/- mice Caput and cauda epididymis sections from *Slc35g3*+/+ and *Slc35g3*-/- mice are comparable.

**Figure supplement 4.** No significant differences in CASA parameters of sperm from *Slc35g3*-/- mice compared to sperm from *Slc35g3*+/+ mice.

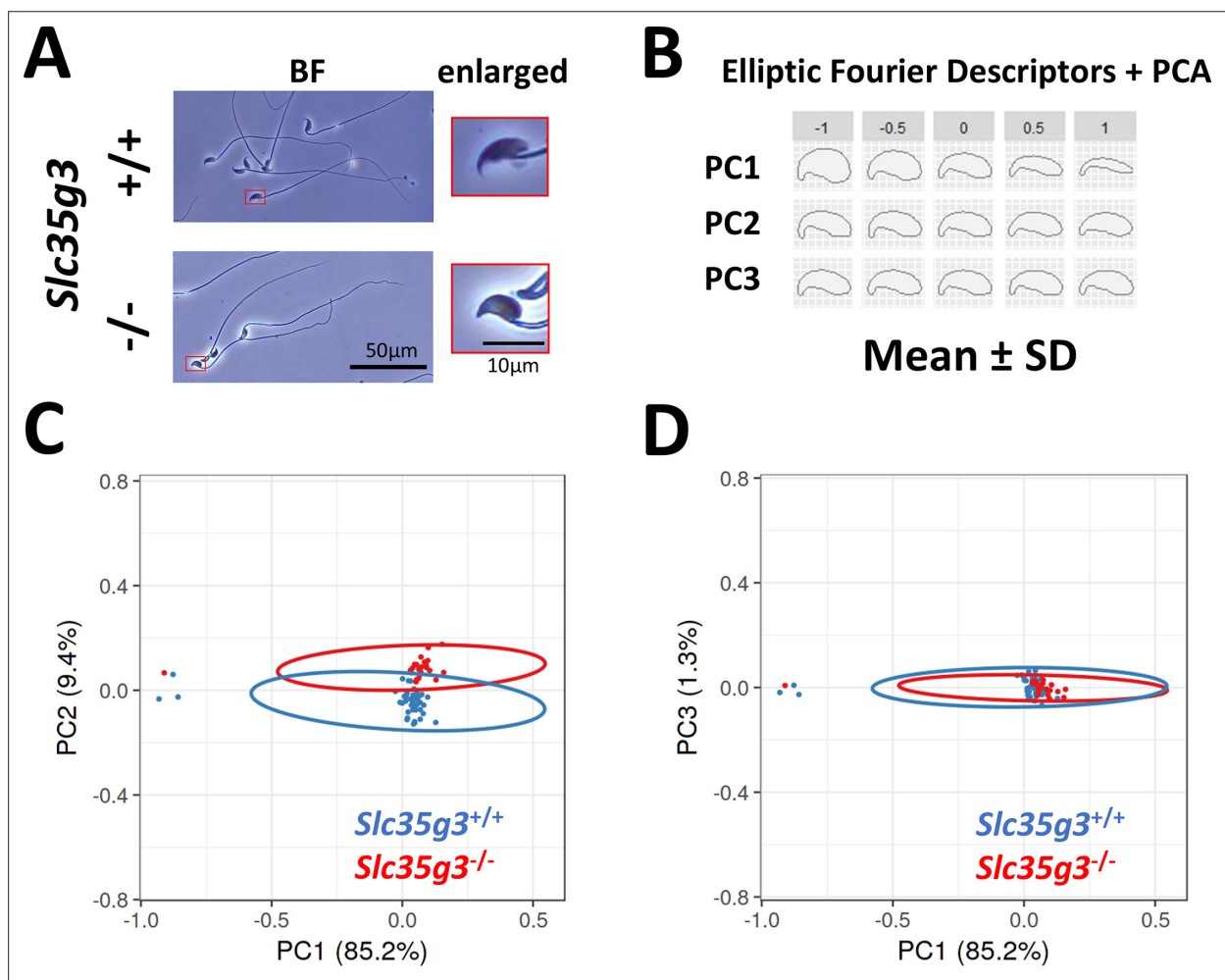

**Figure 3.** *Slc35g3* is involved in the regulation of sperm head morphology. (**A**) Bright-field (BF) views of *Slc35g3*+/+-derived sperm (upper panels) versus *Slc35g3* -/--derived sperm (lower panels); red frames are images enlarged four times. Scale bar: 50 μm for BF images, 10 μm for enlarged ones. (**B**) Morphological characteristics are indicated by mean ± SD of each principal component (PC) following elliptic Fourier analysis; the upper value represents SD, with zero indicating average morphology. (**C, D**) Plots of PC1-PC2 (**C**) and PC1-PC3 (**D**) coordinates of the elliptic Fourier analysis of sperm from *Slc35g3*+/+ mice (blue encircled) versus *Slc35g3*-/- mice (red encircled); circles represent 95% confidence ellipses. Scale bar = 10 μm.

The online version of this article includes the following figure supplement(s) for figure 3:

**Figure supplement 1.** Sperm from *Slc35g3*-/- mice exhibit morphology similar to sperm from *Fam71f2*-/- mice.

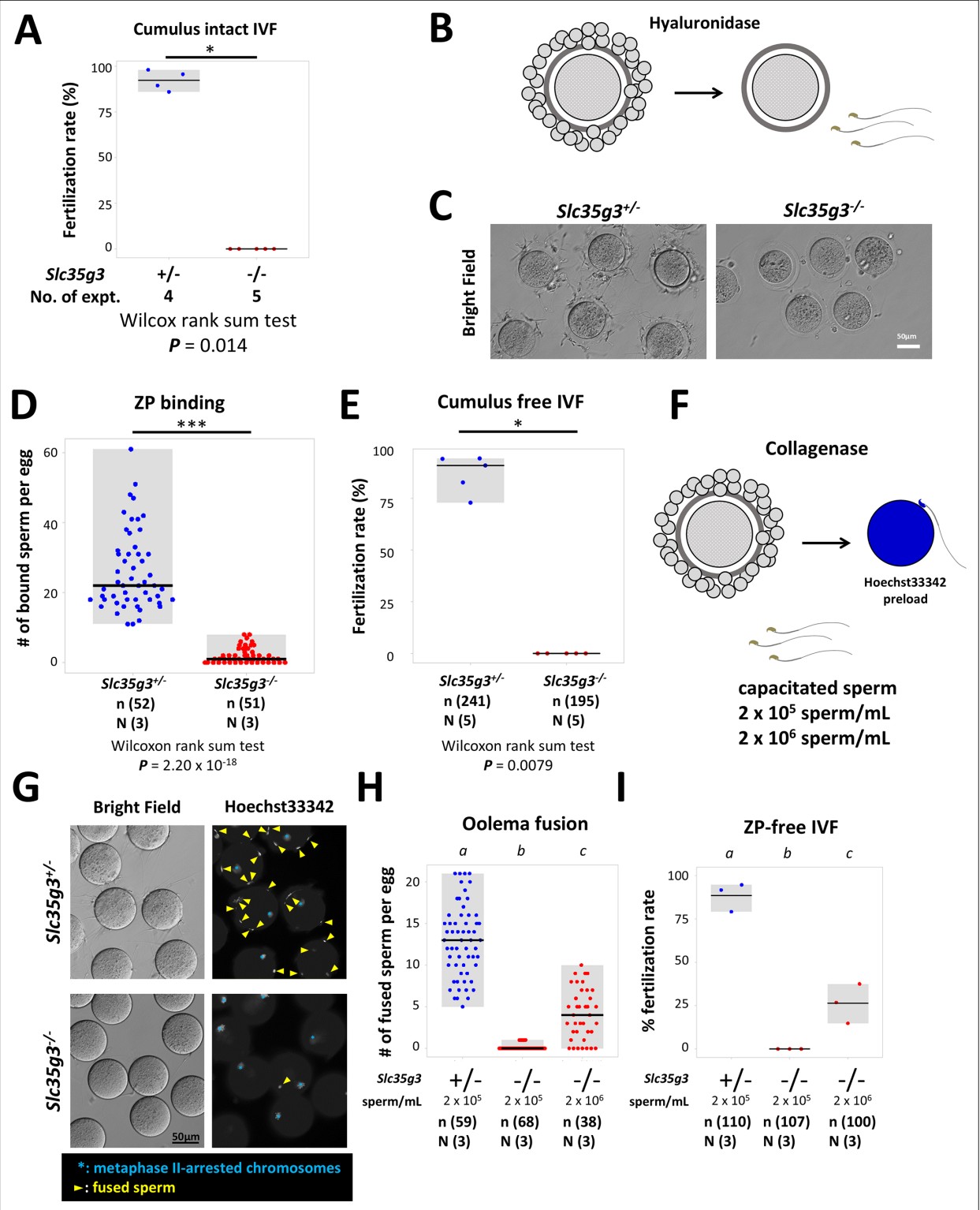

**Figure 4.** *Slc35g3*[−/−]-derived spermatozoa are defective in ZP binding and oolemma fusion. (**A**) The in vitro fertilization (IVF) fertilization rate of cumulus-intact oocytes using *Slc35g3*[+/−] and *Slc35g3*[−/−]-derived sperm. Wilcoxon rank-sum test p=0.014. (**B**) Outline of the procedure of cumulus cell-free IVF. (**C**) *Slc35g3*[+/−]-derived and *Slc35g3*[−/−]-derived sperm binding to cumulus-free oocytes after insemination. Scale bar = 50 μm. (**D**) The number of bound sperm per egg for *Slc35g3*[+/−]-derived and *Slc35g3*[−/−]-derived sperm (Wilcoxon rank-sum test p=2.20 × 10[−18]). (**E**) The fertilization rate of cumulus cell-free IVF using Slc35g3[+/−]-derived and Slc35g3[−/−]-derived sperm. (**F**) The procedure of ZP-free IVF. Wilcoxon rank-sum test; p=0.0079. (**G**) Brightfield and Hoechst33342 staining of oocytes and *Slc35g3*[+/−]-derived and *Slc35g3*[−/−]-derived sperm after insemination into ZP-free oocytes; Yellow arrowheads

*Figure 4 continued on next page*

*Figure 4 continued*

indicate fused spermatozoa and light blue asterisks indicate metaphase II-arrested chromosomes. (**H**) The number of fused sperm per egg using *Slc35g3*[+/-]-derived and *Slc35g3*[-/-]-derived sperm ($2 \times 10^5$ sperm/mL and $2 \times 10^6$ sperm/mL, respectively). Significant differences are indicated by distinct symbols. (**I**) The fertilization rate of ZP-free IVF using *Slc35g3*[+/-]-derived and *Slc35g3*[-/-]-derived sperm ($2 \times 10^5$ sperm/mL and $2 \times 10^6$ sperm/mL, respectively). Significant differences are indicated by distinct symbols.

The online version of this article includes the following figure supplement(s) for figure 4:

**Figure supplement 1.** Oolema fusion observed under high sperm concentration conditions.

**Figure supplement 2.** Pups were produced from spermatozoa from *Slc35g3*[-/-].

control *Slc35g3*[+/-]-derived sperm tail signals marked by red fluorescence were observed within the oviduct (*Figure 5C*). In contrast, *Slc35g3*[-/-]-derived sperm were found in the uterus but not in the oviduct (*Figure 5C*). Thus, the defective UTJ migration is one of the primary causes of *Slc35g3-/-* male infertility.

## *Slc35g3* absence causes a reduced amount and abnormal processing of sperm glycoproteins

To understand the molecular mechanisms behind the disrupted sperm functions of *Slc35g3*[-/-] mice, we analyzed glycoproteins related to each process. First, we examined proteins involved in acrosome formation. Immunoblot analysis of *Slc35g3*[+/+] and *Slc35g3*[-/-] testis lysates showed a reduction in the amount of ZP binding protein 1 (ZPBP1; *Lin et al., 2007*), whereas Golgi-associated PDZ and coiled-coil motif containing (GOPC; *Yao et al., 2002*) levels remained unchanged (*Figure 6A*). Interestingly, SPACA1 (Sperm Acrosome Associated 1; *Fujihara et al., 2012*) exhibited a subtle difference in banding pattern in the *Slc35g3*[-/-] testis lysate. SPACA1 is N-glycosylated, and treatment of testis

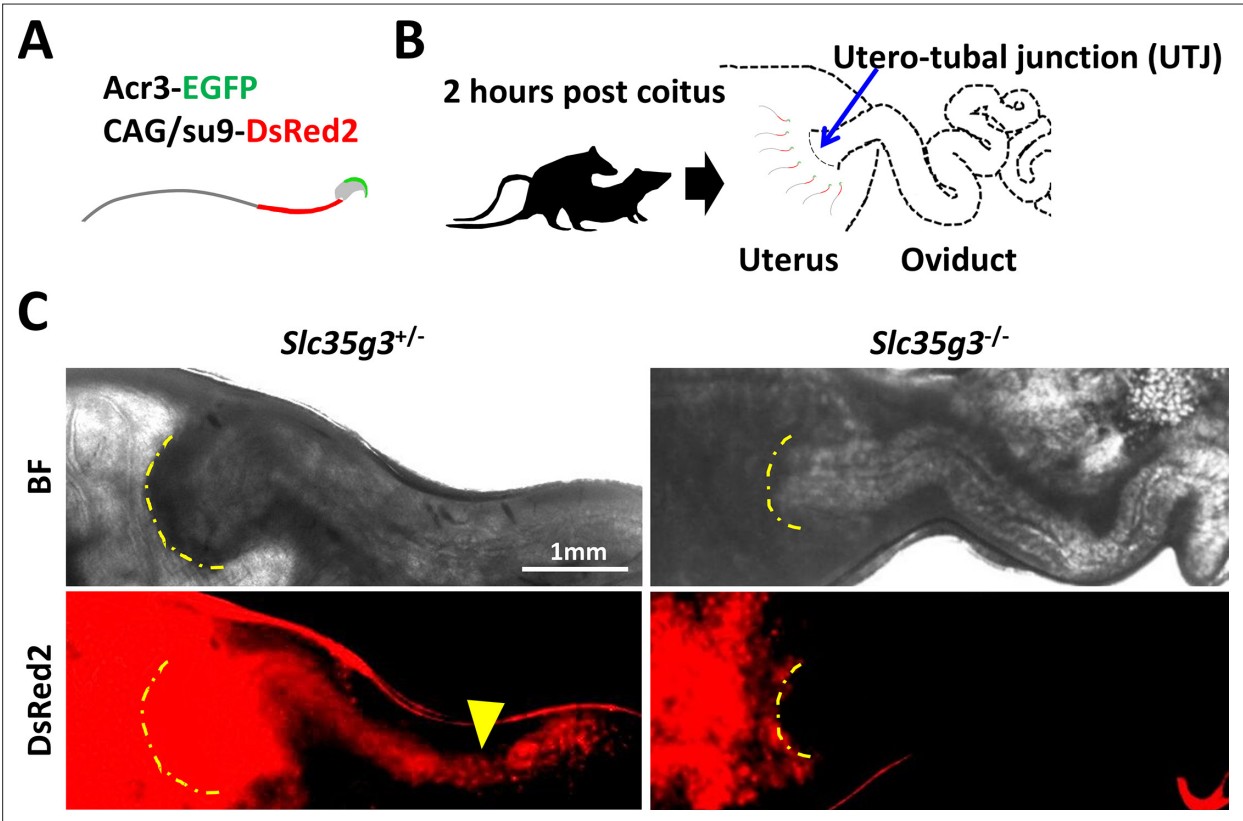

**Figure 5.** *Slc35g3*-deficient mice show impaired sperm migration to the oviduct. (**A**) Illustration of Tg (CAG/su9-DsRed2, Acr3-eGFP) sperm. (**B**) A schematic diagram of the sperm migration assay. (**C**) Bright field (top panel) and Dsred2 (bottom panel) images of the uteri and oviducts of females after mating with control *Slc35g3*[+/-] and *Slc35g3*[-/-] male mice. The yellow dashed line indicates the uterotubal junction (UTJ), and the yellow arrowhead represents the sperm from control *Slc35g3*[+/-] male mice that have traversed the UTJ.

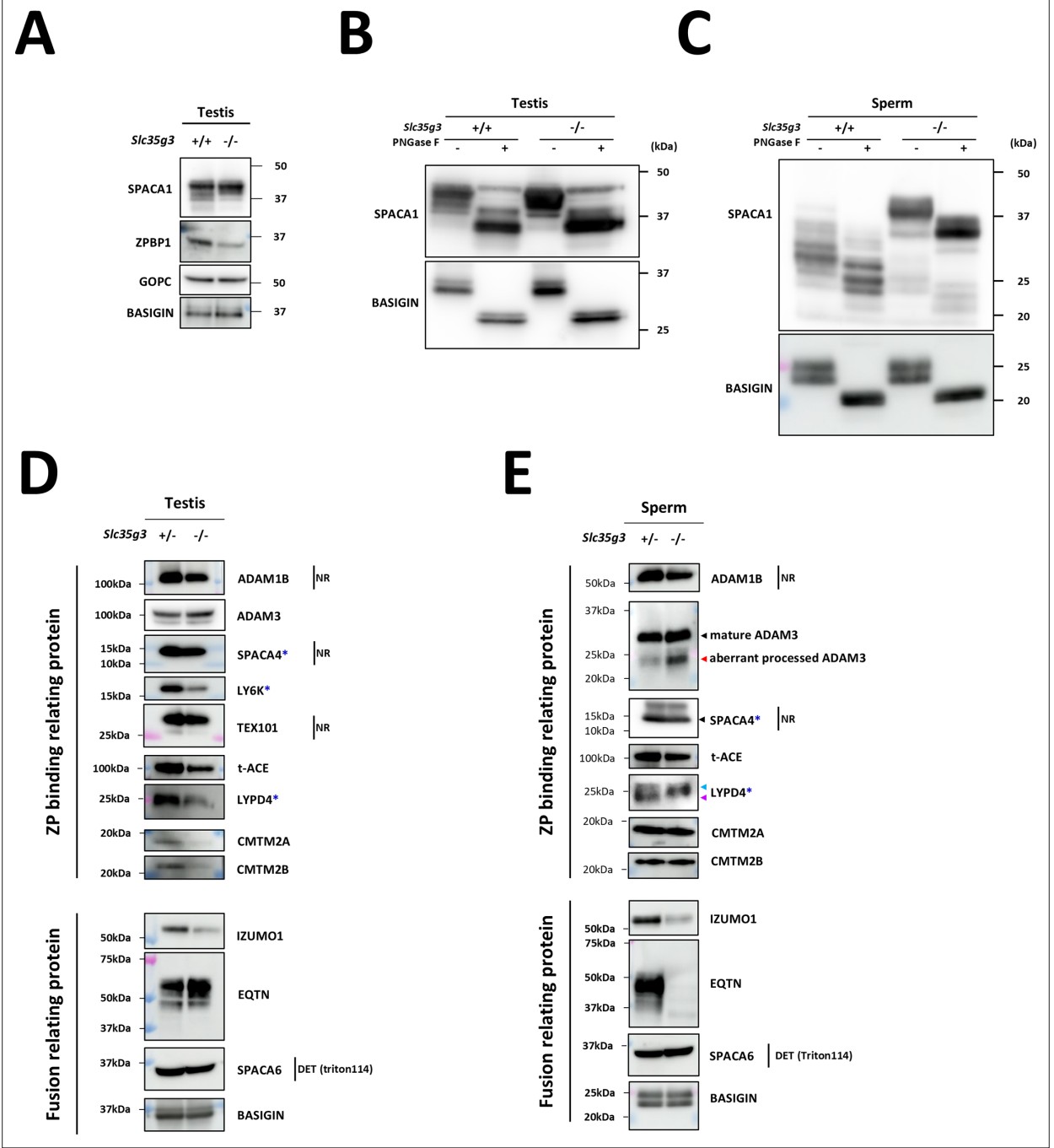

**Figure 6.** Disruption of *Slc35g3* leads to its reduced testicular expression and abnormal processing of multiple sperm proteins. (**A**) Western blot analyses of SPACA1, ZPBP1, and GOPC in *Slc35g3⁺/⁺* and *Slc35g3⁻/⁻* testes, with BASIGIN used as a loading control. (**B**) Western blot analysis of PNGaseF-treated or non-treated SPACA1 in *Slc35g3⁺/⁺* and *Slc35g3⁻/⁻* testes, with BASIGIN used as a loading control. (**C**) Western blot analysis of PNGaseF-treated or non-treated SPACA1 in *Slc35g3⁺/⁺*-derived and *Slc35g3⁻/⁻*-derived spermatozoa, with BASIGIN used as a loading control. (**D**) Western blot analyses of ADAM1B, ADAM3, SPACA4, LY6K, TEX101, t-ACE, LYPD4, CMTM2A, CMTM2B, IZUMO1, EQTN, and SPACA6 in *Slc35g3⁺/⁻* and *Slc35g3⁻/⁻* testes, with BASIGIN used as a loading control. All protein samples were processed under reducing and denaturing conditions unless otherwise specified. Non-reducing and non-denaturing conditions are denoted as NR. For SPACA6 detection, fractions of testis proteins from wild-type and knockout specimens, extracted using Triton X-114, were utilized (abbreviated as DET). Genes marked with blue asterisks show reduced ZP binding upon knockout, whereas ADAM3 remains unaffected. (**E**) Western blot analyses of ADAM1B, ADAM3, SPACA4, t-ACE, LYPD4, CMTM2A, CMTM2B, IZUMO1, EQTN, and SPACA6 in *Slc35g3⁺/⁺*-derived and *Slc35g3⁻/⁻*-derived spermatozoa, BASIGIN used as a loading control. The black arrowhead indicates the predicted protein size, whereas the red arrowhead indicates an aberrantly processed protein isoform. Additionally, the light blue and purple arrowheads mark the two bands observed in the wild-type sample.

*Figure 6 continued on next page*

*Figure 6 continued*

The online version of this article includes the following source data and figure supplement(s) for figure 6:

**Source data 1.** Western blots for *Figure 6*, indicating the relevant bands.

**Source data 2.** Original files for western blot analysis displayed in *Figure 6*.

**Figure supplement 1.** ADAM3 patterns were comparable between Slc35g3$^{+/-}$ and Slc35g3$^{-/-}$ after PNGaseF treatment.

**Figure supplement 1—source data 1.** Western blots for *Figure 6—figure supplement 1*, indicating the relevant bands.

**Figure supplement 1—source data 2.** Original files for western blot analysis displayed in *Figure 6—figure supplement 1*.

**Figure supplement 2.** Distribution of IZUMO1 is expanded in sperm from Slc35g3$^{-/-}$ mice after the acrosome reaction.

and caudal sperm lysates with peptide-N-glycosidase F (PNGase F; *Figure 6B and C*) resulted in comparable SPACA1 band patterns between *Slc35g3$^{+/+}$* and *Slc35g3$^{-/-}$* testes but not in sperm lysates. A similar result was also reported in *Fam71f1* $^{-/-}$ (Garin1b; *Morohoshi et al., 2021*) mice, which exhibit abnormal acrosome formation.

Next, we examined proteins involved in ZP binding. Levels of a disintegrin and metalloprotease (ADAM) 1B (*Kim et al., 2006*) were comparable between *Slc35g3$^{+/-}$* and *Slc35g3$^{-/-}$* in the testis (*Figure 6D*) and sperm (*Figure 6E*). Levels of CKLF-like MARVEL transmembrane domain containing (CMTM) 2A and CMTM2B (*Fujihara et al., 2018b*) were reduced in *Slc35g3$^{-/-}$* testis lysates but not in sperm. The expression pattern of ADAM3 (*Shamsadin et al., 1999*; *Yamaguchi et al., 2009*) was comparable between *Slc35g3$^{+/-}$* and *Slc35g3$^{-/-}$* testis, yet the amount of a smaller isoform was elevated in *Slc35g3$^{-/-}$*-derived sperm lysates, indicating aberrant processing. After PNGaseF treatment of proteins, the ADAM3 band pattern was comparable between *Slc35g3$^{+/-}$* and *Slc35g3$^{-/-}$* (*Figure 6—figure supplement 1*). Given the aberrant ADAM3 band pattern was also observed in testis expressed gene 101 (TEX101) knockout (*Fujihara et al., 2013*) epididymal caput sperm, we examined TEX101 levels through western blot analysis; however, the amount of TEX101 was comparable between *Slc35g3$^{+/-}$* and *Slc35g3$^{-/-}$* testis lysates (*Figure 6D*). Given that the testicular *Ace$^{-/-}$* (t-ACE; *Krege et al., 1995*, *Hagaman et al., 1998*, *Yamaguchi et al., 2006*) caused aberrant localization of ADAM3, we examined t-ACE levels through western blot analysis and found that the amount of t-ACE was comparable between *Slc35g3$^{+/-}$* and *Slc35g3$^{-/-}$* in both testis and sperm lysates (*Figure 6D and E*). In the previous studies, lymphocyte antigen 6 family member K (*Ly6k*)$^{-/-}$ (*Fujihara et al., 2014*), *Spaca4$^{-/-}$* (*Fujihara et al., 2021*), and LY6/PLAUR domain containing 4 (*Lypd4*) $^{-/-}$ (*Fujihara et al., 2019*) sperm showed impaired ZP binding; however, the amount of ADAM3 remained normal. The amount of LY6K was reduced in *Slc35g3$^{-/-}$* testis lysates (*Figure 6D*), and the amount of SPACA4 was comparable between *Slc35g3$^{+/-}$* and *Slc35g3$^{-/-}$* in both testis and sperm lysates (*Figure 6D and E*). However, the amount of LYPD4 in *Slc35g3$^{-/-}$* testis lysates decreased, and the lower molecular weight band disappeared in *Slc35g3$^{-/-}$*-derived sperm lysates (*Figure 6E*), indicating the occurrence of a protein processing error or another non-N-linked oligosaccharide post-translational defect.

Finally, we focused on the inner acrosomal membrane proteins involved in oolemma fusion. IZUMO1 is an N-glycosylated acrosome membrane protein, and the first to be identified as essential for sperm-oolemma fusion using knockout mice (*Inoue et al., 2005*). The levels of IZUMO1 decreased in both the testis and sperm of *Slc35g3$^{-/-}$* mice. Although the amount of IZUMO1 in sperm was less, we did not see any other bands in the western blot analysis. IZUMO1 could relocate to the equatorial segment where fusion occurs after the acrosome reaction in *Slc35g3$^{-/-}$*-derived spermatozoa (*Figure 6—figure supplement 2*). Equatorin (EQTN) is an O-linked glycosylated protein on the inner acrosomal membrane, not essential for oolemma fusion but rather functions in oolemma binding. The EQTN signal showed no difference between *Slc35g3$^{+/-}$* and *Slc35g3$^{-/-}$* testes, but it disappeared in *Slc35g3$^{-/-}$*-derived sperm. Intriguingly, mass spectrometry analysis of sperm lysates showed comparable quantitative values of EQTN between *Slc35g3$^{+/-}$* and *Slc35g3$^{-/-}$* mice (*Supplementary file 1*). With the fact that the anti-EQTN antibody MN9 recognizes both peptide and glycan structures and that the glycan structure (*Toshimori et al., 1992*), our data suggests that EQTN glycosylation is aberrant in *Slc35g3$^{-/-}$*-derived sperm. SPACA6 is known to be lost from all the sperm-oolemma fusion defective sperm (i.e., *Dcst1/2*, *Fimp*, *Izumo1*, *Sof1*, *Spaca6*, and *Tmem95* knockout models) (*Noda et al., 2020*); however, we did not see any difference in the intensity and band pattern using western blot analysis.

## *Slc35g3*⁻/⁻-derived spermatozoa show impaired glycan structures

To analyze the protein glycosylation status during spermatogenesis, we performed lectin blot analyses using testis lysates (*Figure 7A*). The band patterns of concanavalin A (ConA; detecting mannose), *Aleuria aurantia* lectin (AAL; detecting fucose), and *Maackia amurensis* II (MAL-II; detecting sialic acid +core1 structure) were comparable between *Slc35g3*⁺/⁺ and *Slc35g3*⁻/⁻ samples. Notably, with PNA, which detects galactose β1–3 acetyl galactosamine (core 1 structure), the intensity of a band around 60 kDa increased in *Slc35g3*⁻/⁻ testis. As PNA binding is known to be inhibited by any galactose modifications (*Bojar et al., 2022*), the core 1 modifications might be disrupted in the target protein. With *Laetiporus sulphureus* lectin N-terminal domain (LSL-N; detecting LacNAc: galactose-GlcNAc), signal intensities for small proteins decreased. Intriguingly, the difference became evident when we performed lectin blot analysis using mature spermatozoa. Some major signals disappeared in PNA and LSL-N blot analysis (*Figure 7B*).

## Mouse *Slc35g3* overexpression restored glycan levels in HEK293T cells without human *SLC35B4*

To investigate whether SLC35G3 acts as a UDP-GlcNAc transporter, we performed rescue experiments with HEK293T cells. First, we designed two gRNAs to knock out *SLC35B4* encoding a known UDP-N-acetylglucosamine (UDP-GlcNAc) transporter that is highly expressed in HEK293T cells (*Figure 7— figure supplement 1A*). The *SLC35B4* knockout cells were obtained by transfecting pX459 containing two gRNAs and a puromycin-resistant cassette, followed by puromycin treatment (*Figure 7—figure supplement 1B*). After three passages, the cells were transfected with a plasmid expressing *Slc35b2* (encoding a phosphoadenosine phosphosulfate transporter), *Slc35b4*, or *Slc35g3*-mCherry. We found that the introduction of *Slc35b2* did not rescue the amounts of proteins with GlcNAc modification (*Figure 7C*). Conversely, transfection with the *Slc35b4*-expressing plasmid rescued GlcNAc modification levels. Further, the LSL-N signals were rescued by *Slc35g3* transfection (*Figure 7C*), suggesting that SLC35G3 functions as a UDP-GlcNAc transporter.

## T179HfsTer27 and F267L mutants failed to rescue glycan loss in *SLC35B4*-disrupted cells

Among the frameshift mutations found in human genomes (n=76156,, gnomAD; 42), the T179HfsTer27 frame mutation has a relatively high allele frequency ($1.88e^{-3}$), and homozygous mutations are observed in 54 individuals (29 females and 25 males). Moreover, AlphaMissense (*Cheng et al., 2023*), a deep learning model trained on protein sequences and annotations of pathogenicity, predicted 35 pathogenic missense mutations in the *SLC35G3* gene. Among these, two mutations (F215L and F267L) were identified in human genomes, and we focused on F267L, which showed potential detrimental effects according to the evolutionary conservation and protein 3D structure (PolyPhen-2; 44). PCR and subsequent direct sequencing confirmed that the *SLC35G3* expression plasmid (wild-type, FS, or F267L) was introduced into *SLC35B4*⁻/⁻ cells. With lectin blot analysis using LSL-N, the signal decreased by *SLC35B4*⁻/⁻ was rescued by the introduction of wild-type *SLC35G3*, while no signal recovery was observed upon the introduction of FS or F267L (*Figure 7D*). The band patterns of ConA modification remained consistent across all transfected cells. These findings suggest a loss of function in the T179HfsTer27 and F267L mutations.

## Discussion

In this study, we identified SLC35G3 as a testis-specific UDP-GlcNAc transporter to underpin proper sperm glycoprotein synthesis and functions. Although *Slc35g3*⁻/⁻ male mice are viable, healthy, and produce motile sperm, they are completely infertile, revealing a critical and unique role of SLC35G3 for producing functional spermatozoa and male fertility. In vitro studies further implicated latent male infertility due to *SLC35G3* mutations.

During spermatogenesis, *Slc35b4* is expressed in earlier stages, but it decreases and is replaced by *Slc35g3* in later stages (*Figure 1—figure supplement 1*). The reason for the stage-dependent usage of two transporters is unknown, but it may be related to the formation of acrosomes, which are rich in glycosylated proteins, in the later stages of spermatogenesis. Although our in vitro data showed no clear differences (*Figure 7C*), SLC35B4 and SLC35G3 may have different activities and/or functions.

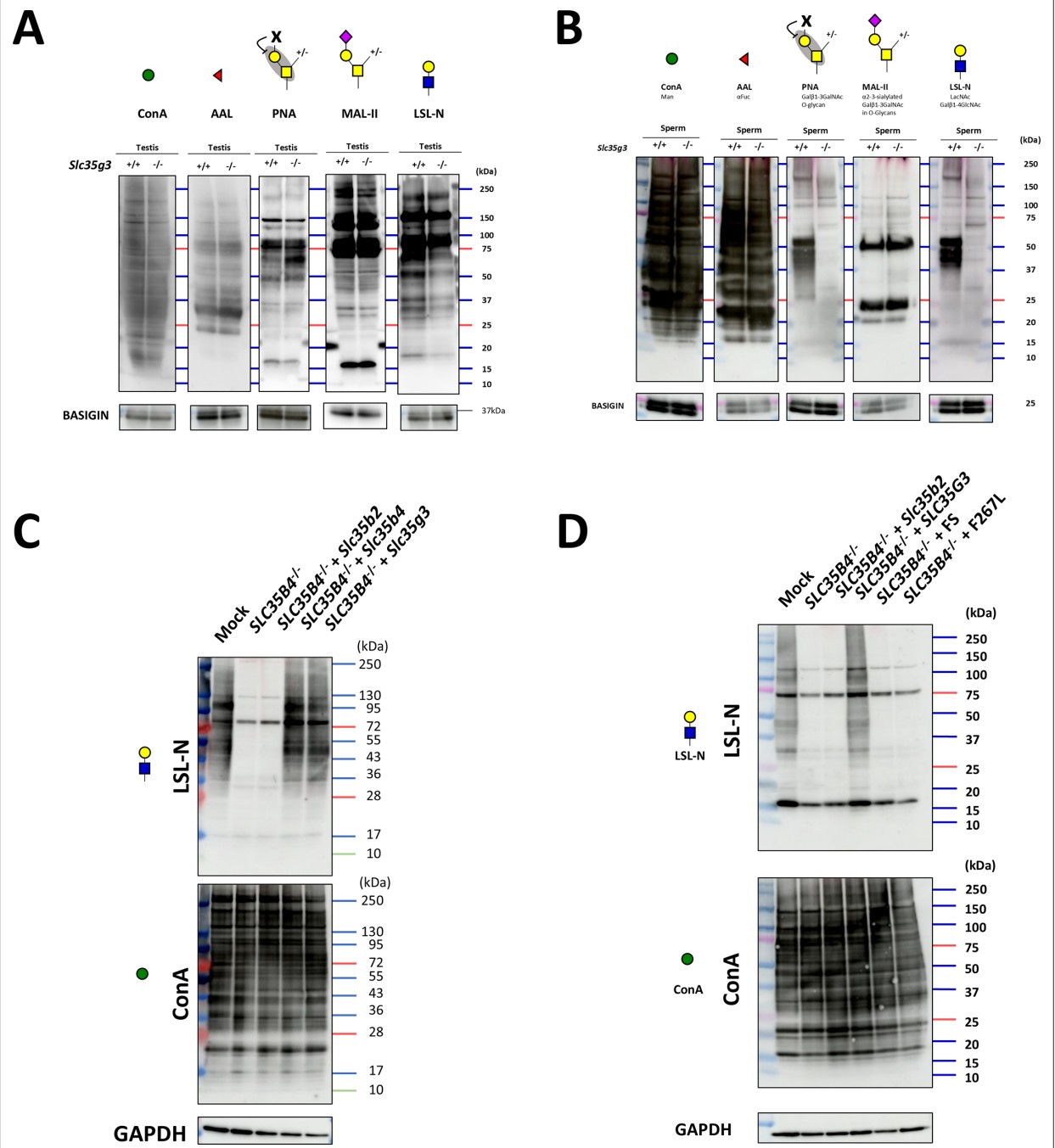

**Figure 7.** *Slc35g3⁻/⁻* testis showed impaired glycan structure. (**A**) Lectin blot (LB) analyses using ConA, AAL, PNA, MAL-II, and LSL-N in *Slc35g3⁺/⁺* and *Slc35g3⁻/⁻* testes, BASIGIN as a loading control. Green circles represent mannose, red triangles fucose, yellow squares GalNAc, yellow circles galactose, purple diamonds sialic acid, and blue squares GlcNAc. (**B**) LB analyses using ConA, AAL, PNA, MAL-II, and LSL-N in *Slc35g3⁺/⁺* and *Slc35g3⁻/⁻* derived spermatozoa. (**C**) LB analyses of LSL-N and ConA in SLC35B4-deficient HEK293T cells, with GAPDH as a loading control. Slc35b2, Slc35b4, and Slc35g3 were expressed in SLC35B4 deficient cells. (**D**) LB analyses of LSL-N and ConA in SLC35G3 mutant transfected SLC35B4 deficient cells, with GAPDH, were used as a loading control. FS: T179HfsTer27.

The online version of this article includes the following source data and figure supplement(s) for figure 7:

**Source data 1.** Western blots for *Figure 7*, indicating the relevant bands.

**Source data 2.** Original files for western blot analysis displayed in *Figure 7*.

**Figure supplement 1.** Design of gRNAs and timeline of the experiment.

Alternatively, SLC35G3 may have a lower optimal temperature because spermatogenesis progresses at a lower temperature (*Bedford, 1991*). Considering the transcript level of *Slc35g3* in later spermatogenesis stages (highest TPM = 320 at round spermatid) is higher than that of *Slc35b4* in the earlier stage (highest TPM = 21 at spermatogonia), spermatids may simply require more UDP-GlcNAc transporter activity. Further in vitro and in vivo studies will be needed to answer these questions, including transgenic mice expressing *Slc35b4* under the *Slc35g3* promoter and vice versa. The answer will also help us to understand why and how spermatogenic cells require a certain number of paralogous genes to be expressed specifically.

Lectin blot analyses revealed no differences in ConA signals targeting terminal mannose (*Figure 7B*), indicating the normal formation of high-mannose-type oligosaccharides for N-glycan biosynthesis in the ER of *Slc35g3*$^{-/-}$ spermatogenic cells (*Aebi, 2013*). For O-glycans, there was an increase in PNA signals (core 1, Gal-GalNAc) and a reduction in LSL-N signals (LacNAc: Gal-GlcNAc), while no changes were observed for MAL-II (sialic acid) and AAL (fucose). These findings suggest that SLC35G3 plays a more important role in glycan elongation rather than core structure, and the impaired elongated glycan structure affected the properties of glycoproteins and following sperm morphology and functions in *Slc35g3*$^{-/-}$ mice.

While spermatogenesis looked grossly normal in *Slc35g3*$^{-/-}$ mice (*Figure 2D, E and G*, *Figure 2—figure supplement 4*), their spermatozoa displayed multiple phenotypic abnormalities in head morphology (*Figure 3*), UTJ migration (*Figure 5*), and fertilization (*Figure 4*). Regarding sperm head malformation, while globozoospermia results in male infertility (e.g., *Zpbp1* [*Lin et al., 2007*], *Gopc* [*Yao et al., 2002*], and *Spaca1* [*Fujihara et al., 2012*] knockout mice), most of the knockout mice with only subtle head malformation can produce offspring, although at lower levels (e.g., *Zpbp2* [*Lin et al., 2007*], *Fam71f2* [*Morohoshi et al., 2021*], and *Garin2-Garin5* [*Wang et al., 2024*] knockout mice). We found a subtle sperm head malformation in *Slc35g3*$^{-/-}$ mice, but it should not be underestimated. For example, it has been shown that mutations in multiple genes synergistically worsen head morphology, even in the heterozygous state (*Martinez et al., 2022*). Although we are still far from unraveling these molecular interactions, we have revealed the importance of SLC35G3-mediated UDP-GlcNAc transport for ZPBP1 stabilization and SPACA1 processing. While these proteins are also found in humans, it is still too early to infer the importance of SLC35G3 in the morphogenesis of human sperm heads. Observing sperm samples from individuals with SLC35G3 mutations would be the most direct approach to address this, and we consider it an important objective for future studies.

*Slc35g3*$^{-/-}$-derived spermatozoa exhibited defective UTJ passage (*Figure 5*) and ZP binding (*Figure 4D*). These defects are commonly linked and observed in many infertile knockout mice, and ADAM3 is absent from most of these knockout spermatozoa (*Fujihara et al., 2019*; *Fujihara et al., 2018a*). However, ADAM3 is present in the *Slc35g3*$^{-/-}$-derived spermatozoa as in four other knockout mouse lines that show the same phenotype (i.e., *Ly6k*, *Pgap1*, *Spaca4*, and *Lypd4* knockout lines). These results suggest that ADAM3 may be dysfunctional in these mutant sperm, or that there may be an unknown factor responsible for UTJ passage and ZP binding. Because LY6K and PGAP1 only function in the testis and disappear from mature spermatozoa, we analyzed the presence of LYPD4 and SPACA4 in mature spermatozoa and found that there was abnormal processing of LYPD4 in *Slc35g3*$^{-/-}$-derived spermatozoa (*Figure 6E*) compared to WT sperm (*Wang et al., 2020*). Since ADAM3 is no longer active in humans, more attention needs to be paid to LYPD4 to understand the sperm-fertilizing ability.

We next focused on the inner acrosomal membrane proteins because *Slc35g3*$^{-/-}$-derived spermatozoa were defective in fusing with oocytes (*Figure 6D and E*). While we did not see any differences in SPACA6 western blot analysis, we found a decrease in IZUMO1 in *Slc35g3*$^{-/-}$-derived spermatozoa, which is consistent with our previous study showing the lack of glycosylation accompanied by a decrease in IZUMO1 levels and a reduction in the number of pups (*Inoue et al., 2008*). Intriguingly, while EQTN was detected by MS analysis (*Supplementary file 1*), signals disappeared in our western blot analysis using an antibody recognizing EQTN O-glycans (*Fujihara et al., 2019*; *Figure 6E*), suggesting the presence of EQTN protein without O-glycans. As *Eqtn* knockout spermatozoa decreased their oolemma binding ability (*Fujihara et al., 2019*), EQTN glycans may directly contribute to oolemma binding. These results suggest that SLC35G3 regulates sperm-oolemma fusion through O-linked glycosylation of inner acrosomal membrane proteins.

Finally, we examined mutations in human *SLC35G3* for their potential risk of male infertility. An in vitro study revealed that the T179HfsTer27 (17–35193772-GT-G) mutation lost sugar-nucleotide transporter activity. According to gnomAD, its frequency is $1.88 \times 10^{-3}$, and 54 individuals have been identified as homozygous. In addition, the observed/expected ratio of single-nucleotide variants causing loss of function was 0.53, suggesting the presence of selective pressure due to mutations. Assessment of their sperm-fertilizing ability would be beneficial to understanding glycosylated protein synthesis and functions in human spermatozoa. Even if the mutation caused male infertility, as we obtained healthy offspring by IVF with higher concentration sperm insemination, intracytoplasmic sperm injection might not be necessary for their treatment.

In conclusion, our research suggests that SLC35G3 functions as a testis-specific UDP-GlcNAc transporter during late spermatogenesis. We reaffirmed that glycosylation-related genes specific to the testis play a crucial role in the synthesis, quality control, and function of glycoproteins on sperm, which are essential for male fertility through their interactions with eggs and the female reproductive system. Furthermore, we demonstrated that human SLC35G3 also exhibits transporter activity and proposed a loss-of-function mutation that may cause male infertility. Further research on this gene and sperm glycoprotein synthesis has the potential to contribute to understanding the causes of male infertility, developing treatments, and advancing contraceptive methods.

## Materials and methods
### Experimental design
In this study, we developed an integrated approach, combining in silico analysis with experimental techniques, to elucidate the functions of SLC35G3. To generate *Slc35g3* knockout (*Slc35g3*$^{-/-}$) male mice, we used the CRISPR/Cas9 system and conducted in silico analysis for off-target/cleavage activity. Male fertility assessment encompasses mating with females, alongside IVF assays. Based on the preliminary literature on the SLC35 family, SLC35G3 is hypothesized to be a nucleotide sugar transporter. Therefore, we performed lectin blot analysis using tissue lysate/HEK293T cell lysate.

### Animals
The article adhered to the ARRIVE guidelines 2.0 for reporting. This study was performed following the standards outlined in the Guide for the Care and Use of Laboratory Animals. All animal experiments were approved by the Animal Care and Use Committee of the Research Institute for Microbial Diseases at Osaka University, Osaka, Japan (#Biken-AP-H30-01). The mice used in the study were sourced from Japan SLC, Inc (Shizuoka, Japan) and were bred under specific pathogen-free conditions. They were housed at 23°C, with a relative humidity of 50%, and a 12 h dark/12 h light cycle, with unrestricted access to water and commercial food pellets ad libitum. All genetically modified mice produced in this study will be accessible through either the RIKEN BioResource Research Center in Ibaraki, Japan, or the Center for Animal Resources and Development (CARD) at Kumamoto University, Japan.

### In silico analysis
Phylogenetic tree analysis was performed using TreeFam (*Ruan et al., 2008*; http://www.treefam.org/), while the Mammalian Reproductive Genetics Database (*Robertson et al., 2020*; https://orit.research.bcm.edu/MRGDv2) was used for mRNA expression analysis of the SLC35 family. Previously reported single-cell RNA sequencing data (*Wang et al., 2023*; https://bis.zju.edu.cn/MCA/) were employed to analyze *Slc35g3* mRNA expression in testicular germ cells. The AlphaFold database (*Jumper et al., 2021*; https://alphafold.ebi.ac.uk/) was utilized for structure prediction, and TOPCONS (*Tsirigos et al., 2015*; https://topcons.cbr.su.se/) was employed for the topological analysis of SLC35G3.

### RNA isolation and reverse transcription polymerase chain reaction
RNA was extracted and purified from various adult tissues of C57BL/6N mice at different stages using TRIzol reagent (Cat. No. 15596018, Thermo Fisher Scientific, Waltham, MA, USA), according to the manufacturer's instructions. Reverse transcription was conducted with the RNA using the SuperScript III First-Strand Synthesis System (Cat. No. 18080051; Thermo Fisher Scientific). PCR amplification was

performed using a KOD Fx Neo (KFX-201; TOYOBO Co., Ltd, Osaka, Japan). The primer sequences used for each gene are listed in *Supplementary file 2*.

## Visualization using fluorescence

Preparation of spermatogenic cells was performed as previously described for testicular cells (*Kotaja et al., 2004*). Briefly, the seminiferous tubules were cut into small pieces, and the contents were extracted by pressing them against a coverslip and frozen. Hoechst33342 (H3570, Thermo Fisher Scientific) and Alexa Fluor 568-conjugated peanut agglutinin (PNA; L32458, Thermo Fisher Scientific) were used to stain the nuclei and acrosomes of cauda epididymal spermatozoa. Observations were performed using a fluorescence microscope (BX53; Olympus, Tokyo, Japan).

## Generation of *Slc35g3* knockout mice

*Slc35g3* knockout mice were generated using the CRISPR/Cas9 system. Guide RNA design and potential off-target analysis were performed using the software programs CRISPRdirect (https://crispr.dbcls.jp/) and CRISPOR (https://crispor.tefor.net/). Fertilized eggs were obtained from the oviducts of super-ovulated B6D2F1 females, which were then mated with BDF1 males. Ribonucleoprotein (RNP) complexes, comprising synthesized CRISPR RNA (crRNA), trans-activating crRNA (tracrRNA), and CAS9 protein, were introduced into fertilized eggs using a NEPA21 super electroporator (Nepa Gene Co., Ltd, Chiba, Japan). The treated eggs were cultured in potassium simplex optimization medium containing amino acids (KSOMaa) until the two-cell stage and were subsequently transplanted into the oviducts of 0.5-day pseudopregnant ICR females. The identity of the pups was confirmed by PCR and Sanger sequencing. Guide RNA and primer sequences are listed in *Supplementary file 2*.

## In vivo male fertility test

Each 8-week-old male, carrying either the *Slc35g3* wild-type or mutated gene, was individually housed with three 8-week-old B6D2F1 female mice for 2 months. Daily observations were made to identify mating plugs, and the number of resultant pups was recorded. A minimum of three males were included in each experimental group for statistical analysis.

## Histological analysis of the testis

Testes were dissected, fixed in Bouin's fluid (Polysciences, Warrington, PA, USA), and embedded in paraffin wax. Subsequently, 5-µm-thick sections were obtained from the paraffin blocks using a Microm HM325 microtome (Microm, Walldorf, DE, Germany). The sections were sequentially dehydrated with xylene and ethanol, followed by a 15 min incubation in a 1% periodic acid solution. After washing under running water for 15 min, the sections were treated with Schiff's reagent (FUJIFILM Wako, Osaka, Japan) for 30 min and then stained with Mayer's hematoxylin solution for 3 min after an additional 15 min wash. Following these processes, the stained samples were observed using SLIDEVIEW VS200 (Olympus).

## Morphological analysis of sperm

Elliptical Fourier transform analysis was performed as previously described (*Kuhl and Giardina, 1982*; *Mashiko et al., 2017*). Briefly, photographs of the spermatozoa were captured using a microscope equipped with a complementary metal oxide semiconductor (CMOS) camera (BX53, DP74, Olympus). The sperm head shape was manually tracked from the photographs, and the elliptic Fourier analysis was performed using Momocs, a contour analysis package of the statistical analysis software R x64 4.1.2 (https://www.r-project.org/). Top PC1-3 scores were visualized using a custom Python code.

## In vitro fertilization

IVF was performed according to the previously established procedures (*Lu et al., 2023*). Cauda epididymal spermatozoa were dispersed in a drop of Toyoda, Yokoyama, Hoshi (TYH) medium (*Toyoda et al., 1971*) covered with paraffin oil (26117-45, Nacalai Tesque Inc, Kyoto, Japan) for 2 h at 37°C under 5% $CO_2$ to facilitate capacitation. Eggs obtained from the oviducts of superovulated females were placed in TYH drops. Cumulus cells were removed by treating the oocytes with 330 µg/mL of hyaluronidase (FUJIFILM Wako Pure Chemical Corp.,) for 5 min. To eliminate the ZP, eggs were treated with 1 mg/mL collagenase (C1639, Merck KGaA, Darmstadt, DE, Germany) for 5 min. The capacitated

spermatozoa were introduced into a drop containing cumulus-intact, cumulus-free, or ZP-free eggs at a final concentration of $2\times10^5$ or $2\times10^6$ spermatozoa/mL. Pronuclei formation was observed 8 h after insemination.

## Computer-assisted sperm analysis

Sperm velocity was analyzed as previously described (*Miyata et al., 2015*). Cauda epididymal spermatozoa were dispersed in 100 µL drops of TYH medium. Sperm motility parameters were measured using the CEROS II sperm analysis system (software version 1.4; Hamilton Thorne Inc, Beverly, MA, USA) at 10 min and 2 h after incubation at 37°C under 5% $CO_2$. More than 200 spermatozoa were analyzed from each male.

## Assessment of sperm passage through the utero-tubal junction

The assay was performed as previously described (*Fujihara et al., 2013*). Briefly, B6D2F1 female mice were subjected to superovulation through intraperitoneal injection of 5 U of equine chorionic gonadotropin (CG), followed by an additional 5 U of human CG (hCG) 48 h later. After 12 h of hCG injection, superovulated females were placed in cages with test males, and vaginal plug formation was monitored at 30 min intervals. Upon confirmation of plug formation, the males were separated from the females. After approximately 2 h of plug formation, the oviducts, along with the connecting portion of the uterus, were excised. These tissues were mounted on slides as whole specimens, covered with coverslips, and examined using fluorescence microscopy (BZ-X810; Keyence Corporation, Osaka, Japan) to assess the presence of sperm containing the mitochondrial DsRed2 marker.

## Plasmid construction

The cDNAs encoding *Slc35g3*, *Slc35b2*, and *Slc35b4* were amplified from mouse testis (C57BL/6N), whereas the cDNA encoding *SLC35G3* was amplified from a human testis cDNA template (Quick Clone#637209, Takara Bio USA Inc, San Jose, CA, USA). The T179HfsTer27 and F267L cDNA mutants were generated using the *SLC35G3* amplicon with the KOD Plus Mutagenesis Kit (SMK-101, TOYOBO Co. Ltd) following the manufacturer's protocol. The *Slc35g3* cDNA was inserted into the mCherry-tagged (C-terminus) pCAG vector, whereas the *Slc35b2*, *Slc35b4*, *SLC35G3*, T179HfsTer27, and F267L cDNAs were cloned into the pCAG vector containing the CAG promoter and rabbit globin poly (A) signal, as previously described (*Niwa et al., 1991*). The primers used to construct these plasmids are listed in *Supplementary file 2*.

## Cell culture and transfection

HEK293T cells (*Tiscornia et al., 2006*) were cultured in DMEM (11995-065, Thermo Fisher Scientific) supplemented with 10% fetal bovine serum (S1560, BioWest, Nuaillé, France) and 1% penicillin-streptomycin-glutamine (10378-016, Thermo Fisher Scientific) at 37°C under 5% $CO_2$. Subsequently, these cells were transiently transfected with the plasmid DNA and cultured.

## Western blot analysis/lectin blot analysis

Immunoblotting procedures closely followed those described previously (*Shimada et al., 2021*). Testis, spermatozoa from the cauda epididymis, and collected cells were immersed in lysis buffer (1% Triton X-100, 50 mM Tris-HCl, pH 7.5, 150 mM NaCl) supplemented with a protease inhibitor cocktail (Cat. No. 25955, Nacalai Tesque Inc) and left to incubate overnight at 4°C. To isolate testicular germ cells (TGC), the testes were minced with a razor blade and passed through a 100 µm nylon mesh. Subsequently, the lysate was centrifuged at 10,000×*g* for 15 min at 4°C. The resulting supernatants were used for either lectin precipitation or SDS-PAGE for immunoblot or lectin blot analysis. PNGase F (P0704S, New England Biolabs Inc, Ipswich, MA, USA) was applied to the testis and sperm lysates to enzymatically treat the glycosidases, following the manufacturer's guidelines.

For lectin blot analysis, a blocking solution (10 mM Tris-HCl, 0.15 M NaCl, 0.05% Tween 20) was employed instead of skim milk for immunoblot analysis. The primary antibody was replaced with biotin-conjugated lectin, and the secondary antibody was substituted with HRP-conjugated streptavidin. The pertinent antibodies and lectins are listed in *Supplementary file 3*.

For lectin precipitation, supernatants from the testis were incubated with lectin-biotin overnight at 4°C, followed by incubation with streptavidin-conjugated Dynabeads (Cat. No. 65001, Thermo Fisher

Scientific) for 1 h at room temperature. After three washes with a mild buffer (42 mM Tris-HCl, pH 7.5, 150 mM NaCl, 0.1% Triton X-100, and 10% glycerol), the complexes were eluted using a sample buffer containing 2-mercaptoethanol.

## Mass spectrometry

The samples were subjected to mass spectrometry analysis as previously described (*Shimada et al., 2021*). The proteins were reduced with 10 mM dithiothreitol (DTT), followed by alkylation with 55 mM iodoacetamide, and digested in-gel by treatment with trypsin and purified with a C18 tip (GL-Science, Tokyo, Japan). The resultant peptides were subjected to nanocapillary reversed-phase LC-MS/MS analysis using a C18 column (25 cm×75 μm, 1.6 μm; IonOpticks, Victoria, Australia) on a nanoLC system (Bruker Daltoniks, Bremen, Germany) connected to a timsTOF Pro mass spectrometer (Bruker Daltoniks) and a modified nano-electrospray ion source (CaptiveSpray; Bruker Daltoniks). The mobile phase consisted of water containing 0.1% formic acid (solvent A) and acetonitrile containing 0.1% formic acid (solvent B). Linear gradient elution was carried out from 2% to 35% solvent B for 18 min at a flow rate of 400 nL/min. The ion spray voltage was set at 1.6 kV in the positive ion mode. Ions were collected in the trapped ion mobility spectrometry (TIMS) device over 100ms, and MS and MS/MS data were acquired over an m/z range of 100–1700. During the collection of MS/MS data, the TIMS cycle was adjusted to 1.1 s and included 1 MS plus 10 parallel accumulation serial fragmentation (PASEF)-MS/MS scans, each containing on average 12 MS/MS spectra (>100 Hz), and nitrogen gas was used as collision gas. Protein identification was carried out using Mascot (version: 2.7.0; Matrix Science, London, UK) regarding Scaffold_4.10.0 (Proteome Software Inc, Portland, OR, USA). Human keratin peptides were excluded from the analysis.

## Statistical analysis

Normality was assessed using the Shapiro–Wilk normality test, and variance was examined using the F-test. Non-parametric tests were performed using the Wilcoxon rank-sum test, whereas parametric tests were performed using the two-tailed Student's *t*-test or Welch's *t*-test. All statistical analyses were performed using R x64 4.1.2 (https://www.r-project.org/). Significance levels were established at *$p < 0.05$, **$p < 0.01$, and ***$p < 0.001$. Data are presented as mean ± s.d. Quantified data were visualized as dot plots using PlotsofData (*Postma and Goedhart, 2019*; https://huygens.science.uva.nl/PlotsOfData/) or custom Python code in Google Colab (https://colab.research.google.com/).

## Data and materials availability

All data needed to evaluate the conclusions in the article are present in the article and/or the supplementary materials. The gene-manipulated mouse lines used in this study were deposited at the RIKEN BioResource Research Center (RIKEN BRC, Tsukuba, Japan) and the Center for Animal Resources and Development (CARD), Kumamoto University (Kumamoto, Japan). These cell lines are available through these centers, subject to scientific review and completion of a material transfer agreement. Requests for access to genetically manipulated mice should be submitted to these centers.

# Acknowledgements

We express our sincere gratitude to Ms. Saki Nishioka and the NPO for Biotechnology Research and Development for their valuable technical support and to the members of both the Department of Experimental Genome Research and Animal Resource Center for Infectious Diseases at the Research Institute for Microbial Diseases (RIMD) of Osaka University, Japan, for their assistance and engaging discussions during the experiments. We acknowledge the RIMD NGS core facility for its valuable support in sequencing and data analysis. We also thank A Ninomiya and F Sugihara for MS analysis (RIMD Core Instrumentation Facility). This work was financially supported by the Ministry of Education, Culture, Sports, Science and Technology/Japan Society for the Promotion of Science (JP21K19569 and JP22H03214 to HM; JP21H05033, JP22H04922, and JP23K20043 to MI); the Japan Science and Technology Agency (JPMJFR211F to HM; JPMJCR21N1 to MI); the Japan Agency for Medical Research and Development (JP23jf0126001, JP23fa627002, and JP23fa627006 to MI); the Takeda Science Foundation (grants to HM and MI); the Eunice Kennedy Shriver National Institute of Child Health and Human Development (R01HD088412 to MM and MI); and the Bill & Melinda Gates Foundation (INV-001902 to MM and MI). OU master plan (JPMXP1323015484 to MI).

# Additional information

## Funding

| Funder | Grant reference number | Author |
|---|---|---|
| Japan Society for the Promotion of Science | JP21K19569 | Haruhiko Miyata |
| Japan Society for the Promotion of Science | JP22H03214 | Haruhiko Miyata |
| Japan Society for the Promotion of Science | JP21H05033 | Masahito Ikawa |
| Japan Society for the Promotion of Science | JP22H04922 | Masahito Ikawa |
| Japan Society for the Promotion of Science | JP23K20043 | Masahito Ikawa |
| Japan Science and Technology Agency | 10.52926/jpmjfr211f | Haruhiko Miyata |
| Japan Science and Technology Agency | 10.52926/jpmjcr21n1 | Masahito Ikawa |
| Japan Agency for Medical Research and Development | JP23jf0126001 | Masahito Ikawa |
| Japan Agency for Medical Research and Development | JP23fa627002 | Masahito Ikawa |
| Japan Agency for Medical Research and Development | JP23fa627006 | Masahito Ikawa |
| Takeda Science Foundation | | Haruhiko Miyata Masahito Ikawa |
| Eunice Kennedy Shriver National Institute of Child Health and Human Development | R01HD088412 | Martin M Matzuk |
| Bill & Melinda Gates Foundation | INV-001902 | Martin M Matzuk |
| Osaka University | OU Master Plan Implementation Project | Masahito Ikawa |

The funders had no role in study design, data collection and interpretation, or the decision to submit the work for publication.

## Author contributions

Daisuke Mashiko, Conceptualization, Data curation, Formal analysis, Methodology, Writing – original draft, Writing – review and editing; Shingo Tonai, Investigation, Writing – review and editing; Haruhiko Miyata, Funding acquisition, Investigation, Visualization, Writing – review and editing; Martin M Matzuk, Conceptualization, Funding acquisition, Visualization, Methodology, Writing – review and editing; Masahito Ikawa, Conceptualization, Supervision, Funding acquisition, Investigation, Methodology, Writing – original draft, Writing – review and editing

## Author ORCIDs

Daisuke Mashiko http://orcid.org/0000-0003-3927-3076
Haruhiko Miyata https://orcid.org/0000-0003-4758-5803
Martin M Matzuk https://orcid.org/0000-0002-1445-8632
Masahito Ikawa https://orcid.org/0000-0001-9859-6217

## Ethics

The manuscript adhered to the ARRIVE guidelines 2.0 for reporting. This study was performed following the standards outlined in the Guide for the Care and Use of Laboratory Animals. All animal experiments were approved by the Animal Care and Use Committee of the Research Institute for Microbial Diseases at Osaka University, Osaka, Japan (#Biken-AP-H30-01). The mice used in the study were sourced from Japan SLC, Inc (Shizuoka, JP) and were bred under specific pathogen-free conditions. They were housed at 23°C, with a relative humidity of 50%, and a 12-h dark/12-h light cycle, with unrestricted access to water and commercial food pellets ad libitum. All genetically modified mice produced in this study will be accessible through either the RIKEN BioResource Research Center in Ibaraki, Japan, or the Center for Animal Resources and Development (CARD) at Kumamoto University, Japan.

Reviewer #2 (Public review): https://doi.org/10.7554/eLife.107494.3.sa1
Author response https://doi.org/10.7554/eLife.107494.3.sa2

---

# Additional files

## Supplementary files

Supplementary file 1. Mass spectrometry data using sperm lysates from *Slc35g3*$^{+/+}$ and *Slc35g3*$^{-/-}$ mice.

Supplementary file 2. Primer and gRNA sequences used in the present studies.

Supplementary file 3. Antibodies used in the present studies.

MDAR checklist

## Data availability

All data needed to evaluate the conclusions in the paper are present in the paper and/or the Supplementary Materials.

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
