## [Editor Report · eLife Assessment]

This **valuable** study reports the physiological function of a putative transmembrane UDP-N-acetylglucosamine transporter called SLC35G3 in spermatogenesis. The conclusion that SLC35G3 is a new and essential factor for male fertility in mice and probably in humans is supported by **convincing** data. This study will be of interest to reproductive biologists and physicians working on male infertility.

---

## [Referee Report · Reviewer #2 (Public review)]

Summary:

This study characterized the function of SLC35G3, a putative transmembrane UDP-N-acetylglucosamine transporter, in spermatogenesis. They showed that SLC35G3 is testis-specific and expressed in round spermatids. Slc35g3-null males were sterile but females were fertile. Slc35g3-null males produced normal sperm count but sperm showed subtle head morphology. Sperm from Slc35g3-null males have defects in uterotubal junction passage, ZP binding, and oocyte fusion. Loss of SLC35G3 causes abnormal processing and glycosylation of a number sperm proteins in testis and sperm. They demonstrated that SLC35G3 functions as a UDP-GlcNAc transporter in cell lines. Two human SLC35G3 variants impaired its transporter activity, implicating these variants in human infertility.

Strengths:

This study is thorough. The mutant phenotype is strong and interesting. The major conclusions are supported by the data. This study demonstrated SLC35G3 as a new and essential factor for male fertility in mice, which is likely conserved in humans.

Weaknesses:

Some data interpretations needed to be revised. These have been adequately addressed in the revised manuscript.

---

## [Author Response]

The following is the authors’ response to the original reviews.

**Reviewer #1 (Public review):**
Summary:In the present manuscript, Mashiko and colleagues describe a novel phenotype associated with deficient SLC35G3, a testis-specific sugar transporter that is important in glycosylation of key proteins in sperm function. The study characterizes a knockout mouse for this gene and the multifaceted male infertility that ensues. The manuscript is well-written and describes novel physiology through a broad set of appropriate assays.Strengths:Robust analysis with detailed functional and molecular assaysWeaknesses:(1) The abstract references reported mutations in human SLC35G3, but this is not discussed or correlated to the murine findings to a sufficient degree in the manuscript. The HEK293T experiments are reasonable and add value, but a more detailed discussion of the clinical phenotype of the known mutations in this gene and whether they are recapitulated in this study (or not) would be beneficial.

Since no patients have been identified, our experiment was conducted to investigate the activity of the mutation found in humans.

(2) Can the authors expand on how this mutation causes such a wide array of phenotypic defects? I am surprised there is a morphological defect, a fertilization defect, and a transit defect. Do the authors believe all of these are present in humans as well?

Thank you for your comment. There are many glycoprotein-coding genes that influence sperm head morphology, fertilization defect, and transit defect have been identified in knockout mouse studies, and most of these are conserved in humans. Therefore, we believe that glycan modification by SLC35G3 is also involved in the regulation of human sperm.

**Reviewer #2 (Public review):**
Summary:This study characterized the function of SLC35G3, a putative transmembrane UDP-N-acetylglucosamine transporter, in spermatogenesis. They showed that SLC35G3 is testis-specific and expressed in round spermatids. Slc35g3-null males were sterile, but females were fertile. Slc35g3-null males produced a normal sperm count, but sperm showed subtle head morphology. Sperm from Slc35g3-null males have defects in uterotubal junction passage, ZP binding, and oocyte fusion. Loss of SLC35G3 causes abnormal processing and glycosylation of a number of sperm proteins in the testis and sperm. They demonstrated that SLC35G3 functions as a UDP-GlcNAc transporter in cell lines. Two human SLC35G3 variants impaired their transporter activity, implicating these variants in human infertility.Strengths:This study is thorough. The mutant phenotype is strong and interesting. The major conclusions are supported by the data. This study demonstrated SLC35G3 as a new and essential factor for male fertility in mice, which is likely conserved in humans.Weaknesses:Some data interpretations need to be revised.

Thank you for comments. We revised interpretations.

**Reviewer #1 (Recommendations for the authors):**
(1) The introduction could be structured more efficiently. Much of what is discussed in the first paragraph appears to be redundant to the second paragraph (or perhaps unrelated to the present manuscript).

In the Introduction, we described the process of glycoprotein formation, (1) quality control or nascent glycoproteins in the ER and its relations importance in sperm fertilizing ability, (2) glycan maturation in the Golgi apparatus and its importance in sperm fertilizing ability, and (3) the supply of nucleotide sugars as the basis of these processes.

We would like to retain this structure in the revised manuscript and appreciate your understanding.

(2) Given the significant difference in morphology between murine and human sperm, can the authors comment on whether these findings are directly translatable to humans?

Thank you for your comment. There are significant differences in sperm morphology between mice and humans, but many glycoprotein-coding genes that influence sperm head morphology have been identified in knockout mouse studies, and most of these are conserved in humans. Therefore, we believe that glycan modification by SLC35G3 is also involved in the regulation of human sperm head morphology. Observing sperm samples from individuals with SLC35G3 mutations is the most direct approach to verify this point and is considered an important goal for future research. The following text has been added to clarify the point:

New Line 338; While these proteins are also found in humans, it is still too early to infer the importance of SLC35G3 in the morphogenesis of human sperm heads. Observing sperm samples from individuals with SLC35G3 mutations would be the most direct approach to address this, and we consider it an important objective for future studies.

(3) Line 194 - while the inability to pass the UTJ may indeed be a component of this infertility phenotype, I would argue that a complete lack of ability to fertilize (even with IVF but not ICSI) suggests that the primary defect is elsewhere. This statement should be removed, and the topic of these two separate mechanisms should be compared/contrasted in the discussion.

We agree that this is an overstatement, so we changed it;

New line 187; Thus, the defective UTJ migration is one of the primary causes of Slc35g3-/- male infertility.

We believe the current statement in the discussion can stay as it is.

Line 379; We reaffirmed that glycosylation-related genes specific to the testis play a crucial role in the synthesis, quality control, and function of glycoproteins on sperm, which are essential for male fertility through their interactions with eggs and the female reproductive system.

(4) Did the authors consider performing TEM to assess the sperm ultrastructure and the acrosome?

Since morphological abnormalities were evident even at the macro level, TEM was not performed in this study. In the future, we plan to use immune-TEM against affected/non-affected glycoproteins when the antibodies become available.

(5) I would argue that Figure 3 should not be labeled as "essential", given the abnormal sperm head morphology compared to humans, the relatively modest difference between the groups on PCA, and more broadly speaking, the relatively poor correlation with morphology and human male infertility. While globozoospermia is clearly an exception, the data in this figure may not translate to human sperm and/or may not be clinically relevant even if it does.

Indeed, other KO spermatozoa with similar morphological features are known to cause a reduction in litter size but do not result in complete infertility. As discussed in line 1, this head shape is not essential for fertilization. Reviewer 2 also pointed out that the phrase "Slc35g3 is essential for sperm head formation" is too strong; therefore, we would like to revise Fig3 title to "Slc35g3 is involved in the regulation of sperm head morphology."

(6) Have the authors generated slc35b4 KO mice?

No, we did not. Since Slc35b4 is expressed throughout the body, a straight knockout may affect other organs or developmental processes. To investigate its role specifically in the testis, it will be necessary to generate a conditional knockout (cKO) model. As this requires considerable cost, time, and labor, we would like to leave it for future investigation.

**Reviewer #2 (Recommendations for the authors):**
(1) Lines 122-123: "it is prominently expressed in the testis, beginning 21 days postpartum (Figure 1B), suggesting expression from the secondary spermatocyte stage to the round spermatid stage in mice." Day 21 indicates the first appearance of round spermatids, but not secondary spermatocytes. Please change to the following: ...suggesting that its expression begins in round spermatids in mice.

I agree with your comment and have revised the text accordingly (New line 114).

(2) Figure 1E: What germ cells are they? The type of germ cells needs to be labelled on the image. Double staining with a germ cell marker would be helpful to distinguish germ cells from testicular somatic cells.

Thank you for your comment. We replaced the Figure 1E as follows.

To distinguish germ cells from testicular somatic cells, we used the germ cell marker TRA98 antibody. Furthermore, based on the nuclear and GM130 staining pattern, we consider that the Golgi apparatus of round spermatids is labeled.

(3) Figure 2C: The most abundant WB band is between 20 and 25 kD and is non-specific. Does the arrow point to the expected SLC35G3 band? There are two minor bands above the main non-specific band. Are both bands specific to SLC35G3? Given the strong non-specific band on WB, how specific is the immunofluorescence signal produced by this antibody? These need to be explained and discussed.

The arrow pointed to the expected size (35kDa).

We thought that these non-specific bands could be due to blood contamination, so we retried with testicular germ cells. We confirmed that non-specific bands disappeared in the subsequent Western blot analysis. The specificity of the immunofluorescence signal is supported by its complete absence in the KO, as shown in the Supplementary Figures. We have decided to include this improved dataset. Thank you for your comment, which helped us improve the data.

**Author response image 1. sa2fig1:** 

(4) Line 184: "Slc35g3-/--derived sperm have defects in ZP binding and oolemma fusion ability, but genomic integrity is intact." Producing viable offspring does not necessarily mean that genomic integrity is intact. Suggestion: Slc35g3-/--derived sperm have defects in ZP binding and oolemma fusion ability but produce viable offspring. Likewise, the Figure S9 caption also needs to be changed.

Thank you for your constructive comment. We have revised the text as you suggested.

(5) Figure 3. "Slc35g3 is essential for sperm head formation". This statement is too strong. It is not essential for sperm head formation. The sperm head is still formed, but shows subtle deformation.

Thank you for your suggestion. We changed as follows:

FIg.3; ”Slc35g3 is involved in the regulation of sperm head morphology.”

(6) Lines 204-205: Figure 6B: "Interestingly, some bands of sperm acrosome-associated 1 (SPACA1; 26) disappeared in Slc35g3-/- testis lysates." I don't see the absence of SPACA1 bands in -/- testis. This needs to be clearly labeled with arrows. On the contrary, the bands are stronger in Slc35g3-/- testis lysates.

Thank you for your comment. After carefully considering your comments, we concluded that using "disappeared" is indeed inappropriate. We would like to revise the sentence as follows: New line 197; "Interestingly, SPACA1 (Sperm Acrosome Associated 1; 26) exhibited a subtle difference in banding pattern in the Slc35g3-/- testis lysate."